# PLUG-AND-PLAY *1.x-Bit* KV CACHE QUANTIZATION FOR VIDEO LARGE LANGUAGE MODELS

## ABSTRACT

Video large language models (VideoLLMs) have demonstrated the capability to process longer video inputs and enable complex reasoning and analysis. However, due to the thousands of visual tokens from the video frames, the key-value (KV) cache can significantly increase memory requirements, becoming a bottleneck for inference speed and memory usage. KV cache quantization is a widely used approach to address this problem. In this paper, we find that 2-bit KV quantization of VideoLLMs can hardly hurt the model performance, while the limit of KV cache quantization in even lower bits has not been investigated. To bridge this gap, we introduce **VidKV**, a plug-and-play KV cache quantization method to compress the KV cache to **lower than 2 bits**. Specifically, (1) for **key**, we propose a mixed-precision quantization strategy in the channel dimension, where we perform 2-bit quantization for anomalous channels and 1-bit quantization combined with FFT for normal channels; (2) for **value**, we implement 1.58-bit quantization while selectively filtering semantically salient visual tokens for targeted preservation, for a better trade-off between precision and model performance. Importantly, our findings suggest that the value cache of VideoLLMs should be quantized in a per-channel fashion instead of the per-token fashion proposed by prior KV cache quantization works for LLMs. Empirically, extensive results with LLaVA-OV-7B and Qwen2.5-VL-7B on six benchmarks show that VidKV effectively compresses the KV cache to 1.5-bit and 1.58-bit precision with almost no performance drop compared to the FP16 counterparts.

## 1 INTRODUCTION

Video large language models (VideoLLMs) have demonstrated strong performance in understanding diverse video contexts (Li et al., 2024e; Lin et al., 2023; Zhang et al., 2023a; Li et al., 2024d; 2023b; Xu et al., 2024; Li et al., 2024b; Wang et al., 2024a; Cheng et al., 2024; Bai et al., 2023; Wang et al., 2024b; Bai et al., 2025). In long video inference scenarios, the key-value (KV) cache stores attention keys and values to avoid redundant computations. However, as the number of video input frames and batch size grows, the substantial memory consumption of the KV cache has emerged as a significant bottleneck in the inference of VideoLLMs, incurring prohibitively large memory usage and slow speed. For instance, in the LLaVA-OV-7B (Li et al., 2024b), with a batch size of 256 and 1,000 input frames, the KV cache required for visual tokens can reach 720 GB[1] by estimation, significantly exceeding the model's own size. Therefore, compressing the KV cache in VideoLLMs is imperative.

In previous works on KV cache compression, most existing approaches focus on removing or merging less critical tokens from the cache to optimize memory usage (Zhang et al., 2023b; Wan et al., 2024; Li et al., 2024f; Ren & Zhu, 2024; Pei et al., 2024; Shen et al., 2024; Tao et al., 2025; Liu et al., 2024a).

---

[1]$(4 \times 28 \times 128 \times 1000 \times 196 \times 256)$ bytes.

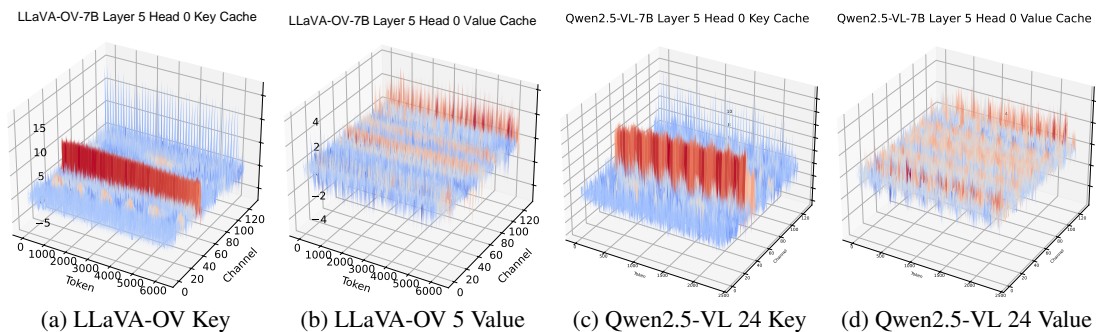

| (a) LLaVA-OV Key | (b) LLaVA-OV 5 Value | (c) Qwen2.5-VL 24 Key | (d) Qwen2.5-VL 24 Value |

Figure 1: Magnitude of KV cache for LLaVA-OV-7B and Qwen2.5-VL-7B. (1) In the key cache, certain channels exhibit significantly large magnitudes, while others display abnormal variations across the channel dimension, making them challenging to quantize. (2) In the value cache, channels exhibit variations in range.

However, such methods may compromise performance as fewer tokens are used. A promising alternative has focused on the quantization of KV cache, a technique that reduces memory usage by converting high-bit floating-point KV caches into lower-bit forms (Liu et al., 2024c; Hooper et al., 2024; Duanmu et al., 2024; Su et al., 2025; Yue et al., 2024; Ashkboos et al., 2024). This group of methods has effectively reduced memory requirements while preserving model performance. However, existing studies have mostly explored this in the context of LLMs. *Its applicability to VideoLLMs remains unexplored, to our best knowledge.*

On VideoLLMs, our preliminary results (as shown in Tab. 1) indicate that, due to the high redundancy of video tokens, the basic group-wise 2-bit KV cache quantization has already achieved promising performance, comparable to the original 16-bit KV cache.

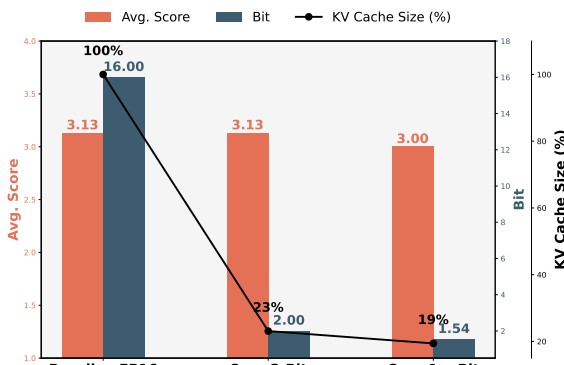

Figure 2: LLaVA-OV-7B model performance with KV cache at FP16 *vs.* 2-bit quantization (Ours) *vs.* 1.5-bit & 1.58-bit quantization (ours). We report the mean scores of two benchmarks, VideoChat-GPT and VideoDC. Empirically, VidKV maintains baseline performance with negligible degradation while reducing the KV cache size by 80%.

This finding suggests the possibility of exploring even lower-bit quantization for KV cache in VideoLLMs. To the best of our knowledge, no prior study has thoroughly analyzed the unique element distribution of KV caches of VideoLLMs in the context of low-bit (1.x bits) quantization. To bridge this gap, we analyze the distribution of KV caches in VideoLLMs. Our analyses suggest that:

- For the key cache, consistent with previous findings (Liu et al., 2024c; Xiao et al., 2023; Lin et al., 2024), certain channels exhibit significantly large magnitudes and substantial variations. These anomalous channels introduce considerable errors in low-bit quantization, leading to model collapse.

- For the value cache, our findings are distinct from prior methods for text LLMs (Liu et al., 2024c), which report substantial *per-channel* magnitude variations, while we find the *per-token* magnitude variations are more obvious (see Fig. 1). This new outlier pattern motivates us to reduce quantization errors in VideoLLMs by adopting a per-channel quantization method for the value cache.

Based on the above analyses, we propose **VidKV**, a lower-bit KV cache quantization method that operates without requiring fine-tuning for VideoLLMs. At its core, we design *1.x-bit* mixed-precision quantization schemes for the key and value caches, respectively. Specifically, **(1) for the key cache**, we employ a straightforward yet effective range-based channel evaluation method to perform 2-bit quantization on anomalous

channels and 1-bit quantization on normal channels. Our findings indicate that transforming the key cache to the frequency domain via the Fast Fourier Transform (FFT) not only stabilizes the distribution of elements across channels but also mitigates the impact of outliers, thereby enhancing quantization accuracy and reducing the complexity of the quantization process. Consequently, we convert the key cache from the time domain to the frequency domain before performing 1-bit quantization and subsequently restore it to the time domain using the inverse FFT (IFFT). **(2) For the value cache**, we implement *1.58-bit* quantization, mapping the values to the set $\{-1, 0, 1\}$, which can bring benefits with proper implementations (Ma et al., 2024; Yang et al., 2024). The matrix multiplication between the value and attention weight can be reformulated as an addition operation, thereby reducing computational energy consumption. In addition, as an option to better trade off precision and model performance, we introduce a token protection mechanism to identify a small set of critical tokens based on their semantic relevance; tokens in this subset are preserved at 2-bit precision during value cache quantization. By doing so, the performance can be significantly preserved. Notably, KV cache quantization in VideoLLMs is essential to mitigate memory and computational bottlenecks. As shown in Fig. 2, VidKV maintains FP16 performance with negligible degradation while reducing the KV cache size by 80%, and using per-channel quantization for value cache can be lossless at 2-bit precision.

Our contributions in this work are summarized as follows:

- We introduce a training-free *plug-and-play 1.x-bit* KV cache quantization framework tailored for video LLMs, *for the first time*. Leveraging distribution characteristics, the method features mixed-precision quantization schemes designed separately for key and value caches.

- For key cache, we propose a simple yet effective range-based way to split the channels into anomalous and normal ones, and then quantize the anomalous channels to 2 bits, and normal channels to 1 bit in the frequency domain.

- For value cache, we propose a 1.58-bit quantization scheme while selecting a few semantically salient tokens for protection, offering an option to better trade off performance with precision. Importantly, we find that in contrast to previous LLM studies, the value cache of VideoLLMs is more suitable for *per-channel* quantization.

- Experimental results on several benchmarks show that VidKV effectively compresses the KV cache to 1.5-bit and 1.58-bit precision, with almost no accuracy drop compared to the FP16 counterparts.

## 2 RELATED WORK

### 2.1 VIDEO LARGE LANGUAGE MODELS

With the rapid blooming of large language models (LLMs) and multimodal large language models (MLLMs), many works have explored incorporating video encoders and LLMs (termed as VideoLLMs) for the video understanding and reasoning tasks (Lin et al., 2023; Ataallah et al., 2024; Maaz et al., 2023; Jin et al., 2024b; Luo et al., 2023; Wang et al., 2024a; Li et al., 2024c;c; Jin et al., 2024a). Regardless of good performance, the efficiency of VideoLLMs is usually limited due to large amount of frames in the videos. Improving efficiency has been a focus in recent VideoLLM works (Shao et al., 2025; Liu et al., 2025; Shen et al., 2025). For example, VideoLLaMA (Zhang et al., 2023a) utilized a Q-Former module (Li et al., 2023a) to pool the video tokens. Xgen-MM-Vid (Ryoo et al., 2024) learns a compact video representation with only 32 tokens. MovieChat (Song et al., 2024) introduced a memory module to merge and store the video tokens. Although the potential of VideoLLMs for video understanding and inference is increasingly recognized, the tens of thousands of visual tags required for long videos significantly increase the KV cache size, thereby affecting inference time and memory requirements. Consequently, different from previous works (Tao et al., 2025; Huang et al., 2024), we explore the lower-bit quantization for VideoLLMs KV caches for the first time.

Figure 3: Overview of our proposed method VidKV. We implement *1.x-bit* mixed-precision quantization for the key cache and 1.58-bit quantization for the value cache. In addition, as shown in the figure, to balance precision and model performance, we protect important visual tokens in the value cache. It is noteworthy that we perform mixed-precision quantization on the key cache along the channel dimension, whereas on the value cache, we apply mixed-precision quantization along the token dimension. All key-value caches are quantized in a *per-channel* fashion, different from prior KV cache quantization methods for LLMs such as KIVI (Liu et al., 2024c).

## 2.2 KV CACHE QUANTIZATION

KV cache quantization optimizes the storage of pre-computed keys and values, alleviating the memory bottleneck by reducing memory consumption and accelerating generation (Liu et al., 2024c; Hooper et al., 2024; Duanmu et al., 2024; Su et al., 2025; Yue et al., 2024; He et al., 2024; Zhang et al., 2024b). KVQuant (Hooper et al., 2024) introduces sensitivity-based and dense-and-sparse quantization techniques for the KV cache, aiming to minimize quantization errors. KIVI (Liu et al., 2024c) analyzes the distribution differences between keys and values in the Multi-Head Attention module. Based on the observations, they quantize keys per-channel and values per-token using group-wise quantization into INT2 while retaining the most recent window in FP16. CQ (Zhang et al., 2024b) proposes to couple multiple key and value channels together for quantization to exploit their dependency. Unlike existing works that primarily focus on LLMs, we aim to analyze and explore the unique characteristics of the KV cache in VideoLLMs, which contain both temporal and spatial features from the video modality.

## 3 PRELIMINARIES

### 3.1 BACKGROUND ON VIDEO LLM INFERENCE

Video LLM inference typically comprises two stages: *prefilling* and *decoding*.

**(1) Prefilling Stage.** During the prefilling phase, the model processes the token sequence generated from the prompt and produces the initial output token, while each attention layer computes and stores KV pairs. Let $\mathbf{X_s} \in \mathbb{R}^{l_s \times d}$, $\mathbf{X_v} \in \mathbb{R}^{l_v \times d}$, and $\mathbf{X_t} \in \mathbb{R}^{l_t \times d}$ denote the system token, visual token, and text token, respectively, where $l_s$, $l_v$, and $l_t$ represent their corresponding input token lengths, and $D$ is the hidden dimension of the model. In each layer, the KV cache is derived as follows:

$$K = \mathbf{X} \cdot W_k, \quad V = \mathbf{X} \cdot W_v, \quad \mathbf{K_{cache}} \leftarrow K, \quad \mathbf{V_{cache}} \leftarrow V, \tag{1}$$

where $\mathbf{X} = \text{concat}[\mathbf{X_s}, \mathbf{X_v}, \mathbf{X_t}]$ and $W_k, W_v \in \mathbb{R}^{d \times d}$ are the weight matrices.

**(2) Decoding Stage.** In the decoding phase, owing to the KV cache, the model takes a single token $x \in \mathbb{R}^{1 \times d}$ as input. Subsequently, the attention output $\mathbf{A}$ can be calculated as

$$Q_x = x \cdot W_q, \quad K_x = x \cdot W_k, \quad V_x = x \cdot W_v, \tag{2}$$

$$K \leftarrow [\mathbf{K_{cache}}, K_X], \quad V \leftarrow [\mathbf{V_{cache}}, V_x], \quad \mathbf{A} = \text{Softmax}\left(\frac{Q_x(K)^\top}{\sqrt{D}}\right) V. \tag{3}$$

Table 1: Results of simulated KV cache quantization under various configurations. $\mathbb{C}$ denotes per-channel quantization, while $\mathbb{T}$ represents per-token quantization. The quantization range for 1.58-bit quantization is $\{-1, 0, 1\}$. *Range*, *Variance*, and *Outlier* are the metrics employed for channel selection in the mixed-precision quantization of the key cache, where *Range* is defined as $max - min$.

| LLaVA-OV-7B | Bit (K/V) | VideoDC | MovieChat |
|---|---|---|---|
| Baseline | 16 | 3.01 | 47.87 |
| K - $\mathbb{C}$, V - $\mathbb{C}$ | 2 / 2 | **3.03** | **47.68** |
| K - $\mathbb{C}$, V - $\mathbb{T}$ | 2 / 2 | 3.00 | 43.63 |
| K - $\mathbb{C}$, V - $\mathbb{C}$ | 1.5 / 1.58 | **2.79** | **47.08** |
| K - $\mathbb{C}$, V - $\mathbb{T}$ | 1.5 / 1.58 | 2.21 | 13.76 |
| Variance | 1.5 / 2 | 2.71 | 45.11 |
| Range | 1.5 / 2 | **2.95** | **48.28** |
| Outlier | 1.5 / 2 | 2.51 | 32.87 |

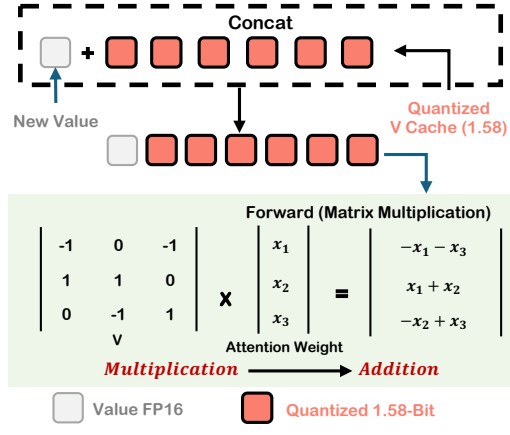

Figure 4: Illustration of our 1.58-bit quantization for the value cache during the decoding stage.

## 3.2 KV Cache Quantization

The n-bit integer KV cache quantization and dequantization process is formulated as follows:

$$Q(\mathbf{X}) = \text{clamp}\left(\left\lfloor \frac{\mathbf{X} - z_{\mathbf{X}}}{s_{\mathbf{X}}} \right\rceil, \, 0, \, 2^n - 1\right), \quad \mathbf{X}' = Q(\mathbf{X}) \cdot s_{\mathbf{X}} + z_{\mathbf{X}}, \tag{4}$$

where $s_{\mathbf{X}} = \frac{\max(\mathbf{X}) - \min(\mathbf{X})}{2^n - 1}$ is the scaling factor, $z_{\mathbf{X}} = \min(\mathbf{X})$, and $\lfloor \cdot \rceil$ indicates round operation.

Notably, for video LLMs, the basic 2-bit KV cache quantization is sufficient to preserve model performance due to the high redundancy of video tokens. This observation motivated us to investigate the lower-bit KV cache quantization video LLMs.

## 4 Methodology

In Sec. 4.1, we are about to analyze the distribution characteristics of KV caches in video LLMs, and our observations indicate that 2-bit quantization can hardly hurt the model performance due to the significant visual token redundancy, leading us to explore even lower-bit quantization. Based on these findings, we shall present VidKV, our *1.x-bit* KV cache quantization method for video LLMs, as detailed in Secs. 4.2 and 4.3.

## 4.1 KV Cache Distribution of video LLMs

Previous studies have examined KV cache distributions in LLMs, but these findings have not been fully validated for video LLMs. As shown in Fig. 1, the key cache often contains outlier channels with significantly larger amplitudes, consistent with prior work Liu et al. (2024c); Hooper et al. (2024). Such abnormal variations complicate quantization, motivating our mixed-precision approach: applying lower-bit quantization to stable channels while reserving higher precision for anomalous ones. In contrast, the value cache is more stable across channels but varies along the token dimension. This makes per-channel quantization more effective than per-token quantization—contrary to prior LLM observations (Liu et al., 2024c). As confirmed in Tab. 1, per-channel quantization achieves higher accuracy for value caches, even with 2-bit settings, while remaining nearly lossless.

## 4.2 MIXED-PRECISION QUANTIZATION FOR KEY CACHE

**(1) Channel Selection.** As analyzed, the key cache contains certain anomalous channels that pose challenges for lower-bit quantization. To address this, we explore a mixed-precision quantization approach. Specifically, we first assess the quantization difficulty of each channel. Channels that are easier to quantize (normal) undergo 1-bit quantization, while more abnormal channels are assigned 2-bit quantization to minimize error.

Thus, properly splitting the channels into abnormal and normal groups is a critical problem here. It is known that quantization becomes increasingly challenging to assess when magnitude distributions exhibit drastic fluctuations and contain numerous outliers. Therefore, we explored several evaluation methods along the channel dimension, including *Variance* $\sigma^2$, *Range* $R = \max(K) - \min(K)$, and the number of *Outliers* $N_{\text{outliers}} = \sum_{i=1}^{l} \mathbb{I}(K_i > M \cdot \bar{K})$, where $C$ is the number of tokens, $\mathbb{I}(\cdot)$ is an indicator function that returns 1 if the condition is met, otherwise 0 and $M$ is a predefined threshold.

Tab. 1 (marked in light blue background) presents the results of an average 1.5-bit quantization (where 50% of the channels undergo 1-bit quantization) for the key cache using different evaluation methods. For all configurations, we set the group size to 32, $M = 3$, and maintain the value cache at a fixed 2-bit quantization. Specifically, we observe that evaluating anomalous channels using the *Range* achieves near-lossless quantization accuracy, whereas the other two methods exhibit a certain degree of performance degradation. As shown in Fig. 3, we select $k\%$ abnormal channels in the key for 2-bit quantization by evaluating the range in each channel, while assigning the remaining normal channels to 1-bit quantization.

**(2) FFT-based 1-Bit Quantization.** As analyzed in Sec. 4.1, the key cache contains numerous abnormal channels, and the distribution of each channel in the time domain exhibits sharp fluctuations, which not only complicates 1-bit quantization but also results in the uneven accumulation of quantization errors across different channels. To address this, we propose to apply *Fast Fourier Transform* (FFT) to transform data from the time domain to the frequency domain and mitigate large oscillations in the channel dimension by leveraging frequency domain properties (such as increased stability and energy concentration), as shown in Fig. 5. FFT is widely used for outlier smoothing (Tseng et al., 2024), and it does not add significant computational overhead (less than 5%).

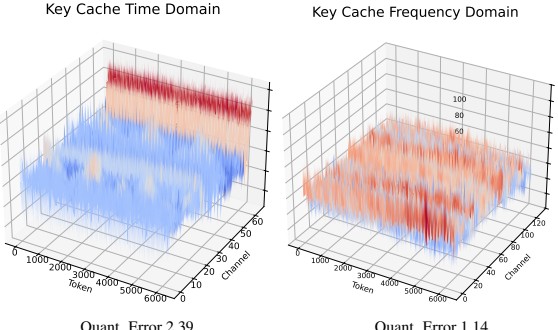

Figure 5: Analysis of the normal channel of key cache shows that FFT transformation smooths the frequency-domain distribution, reducing quantization error.

Due to the significant reduction in quantization error, 1-bit quantization not only substantially decreases storage overhead but also mitigates the loss of effective information. The specific quantization and dequantization process can be written as follows,

$$Q(\mathbf{X}_{\text{fft}}) = \text{sign}\big(\text{FFT}(\mathbf{X})\big) + 1 \in \{0, 1\}, \quad \mathbf{X}' = \text{IFFT}[Q(\mathbf{X}_{\text{fft}}) \cdot s_{\text{fft}} + z_{\text{fft}}], \qquad (5)$$

where the scale $s_{\text{fft}} = \text{Mean}(|\text{FFT}(\mathbf{X})|)$ and the zero offset $z_{\text{fft}} = 0$.

## 4.3 1.X-BIT QUANTIZATION FOR VALUE CACHE

**(1) 1.58-Bit Quantization.** For the value cache, we propose to employ a promising lower-bit quantization approach: 1.58-bit quantization as the base scheme. While 1.58-bit quantization has previously demonstrated its effectiveness for LLM *weight* quantization (Ma et al., 2024), we explore its applicability to KV cache quantization for the *first* time.

The 1.58-bit means ternary quantization, *i.e.*, mapping a value to $\{-1, 0, 1\}$. Concretely, in the prefilling stage, we compute the average value as the threshold and subsequently constrain the values to -1, 0, or +1:

$$Q(V)_{1.58} = \text{sgn}(V) \cdot \mathbf{1}_{|V|>\alpha} \tag{6}$$

where $\alpha = \gamma \cdot \text{mean}|V|$, and $\gamma$ is a hyperparameter. Notably, a significant advantage of the 1.58-bit quantization is its potential for faster and cheaper computing. As shown in Fig. 4, the matrix multiplication between value and attention weights can be replaced with addition and subtraction, significantly reducing the computational energy consumption. Although 1-bit quantization of the value cache still poses challenges, the 1.58-bit scheme retains its advantages, especially its computational efficiency.

**(2) Semantic Token Protection (STP).** Additionally, in video LLMs, certain visual tokens play a more crucial role in the inference process due to their strong correlation with the input text, inspiring us to apply higher protection to these critical visual tokens through 2-bit quantization, thereby minimizing quantization errors for these essential tokens. Furthermore, this approach ensures that lower-bit quantization of other tokens does not adversely impact the accuracy of critical tokens. As illustrated in Fig. 3, the selection mechanism relies on cross-modal attention scores between each vision token and the text query:

$$\mathcal{I} = X_v(i) \cdot X_t(j)^\top. \tag{7}$$

Selective application of 2-bit quantization to the top $n$ visual tokens preserves the semantic integrity of the most informative visual features, where $n = p \cdot l_v$ and $p$ is the percentage of tokens protection. Meanwhile, the remaining tokens undergo 1.58-bit quantization, maintaining resource efficiency while preserving essential semantic information.

## 5 EXPERIMENTAL RESULTS

### 5.1 EXPERIMENT SETTINGS

**Models.** We select two of the most widely used video large language model families to evaluate our VidKV: LLaVA-OneVision (Li et al., 2024b) and Qwen2.5-VL (Bai et al., 2025). We utilize the Hugging Face Transformers codebase and implement our VidKV algorithm on top of it. Specifically, we evaluate LLaVA-OneVision-7B on 8 RTX 4090 GPUs, supporting up to 32 video input frames, and Qwen2.5-VL-7B on 8 A6000 GPUs, supporting up to 16 input frames.

**Tasks.** For the evaluation of VLLMs, we do not select common video question-answering (QA) tasks where only a single word is generated. Instead, we adopt the VideoDC (LMMs-Lab, 2024), VideoChat-GPT (Maaz et al., 2023), MovieChat (Song et al., 2024), TempCompass (Liu et al., 2024b), VATEX (Wang et al., 2019), and WorldQA (Zhang et al., 2024c) benchmarks to evaluate long text generation performance.

**Implementation Details.** In group-wise quantization, we set a residual length inspired by KIVI (Liu et al., 2024c) to store the parts that are not divisible. We set the quantized group size $G$ to 32 and the residual key-value cache length $R$ to 128 in all experiments. The hyperparameter for threshold calculation in 1.58-bit quantization $\gamma$ is set to 0.7 for 1.58-bit quantization. The key cache is quantized at mixed precisions, ranging from 1-bit to 2-bit. Owing to FFT computations, FFT-based 1-bit quantization is applied exclusively to key-1.5-bit ($k = 0.5$) and key-1.75-bit ($k = 0.75$), while standard 1-bit quantization is used otherwise. All benchmarks utilize the LMMs-Eval (Zhang et al., 2024a; Li et al., 2024a) framework for evaluation, and all evaluated code remains consistent with the official implementation.

### 5.2 MAIN RESULTS AND ANALYSES

This section presents the primary results of cache quantization for *1.x-bit* KV representations. Notably, most existing methods focus on 2-bit KV-cache quantization for text-only LLMs. We present, for the first time,

Table 2: Results of different methods and quantization settings. For all values, higher is better. The best result of each metric in each model is in **bold**, and the second best is underlined. 1.66-bit means 20% tokens for 2-bit and 80% tokens for 1.58-bit

| Method | Settings | | VideoDC | TempCompass | VideoChat-GPT | | | | | | Moviechat | | WorldQA |
|---|---|---|---|---|---|---|---|---|---|---|---|---|---|
| | K-(Bit) | V-(Bit) | GPT Sco. | Avg. | CI | DO | CU | TU | CO | Avg. | GPT Score | Acc. | GPT Sco. |
| LLaVA-OV-7B | | | | | | | | | | | | | |
| Baseline | 16-Bit | | 3.01 | 49.05 | 3.47 | 2.97 | 3.71 | 2.74 | 3.49 | 3.27 | 3.09 | 47.87 | 0.328 |
| KIVI | 2-Bit (K-ℂ V-𝕋) | | 3.00 | 49.70 | 3.48 | **2.95** | 3.68 | **2.72** | 3.35 | 3.24 | 3.05 | 46.63 | 0.326 |
| VidKV | 2-Bit (K-ℂ V-ℂ) | | **3.03** | **50.69** | 3.48 | **2.95** | **3.69** | **2.72** | **3.55** | **3.27** | 3.08 | 47.68 | **0.327** |
| VidKV | 1.50 | 2.00 | 2.95 | 50.45 | **3.49** | 2.94 | 3.63 | 2.70 | 3.38 | 3.23 | **3.12** | **48.28** | 0.322 |
| | 1.50 | 1.58 | 2.79 | 47.35 | 3.32 | 2.77 | 3.57 | 2.58 | 3.10 | 3.06 | 3.11 | 47.08 | 0.313 |
| | 1.25 | 1.58 | 2.53 | 45.21 | 3.29 | 2.66 | 3.59 | 2.47 | 3.06 | 3.01 | 3.06 | 47.21 | 0.309 |
| VidKV ($p = 0.2$) | 1.50 | 1.66 | 2.89 | 47.55 | 3.35 | 2.79 | 3.60 | 2.66 | 3.11 | 3.10 | 3.11 | 47.25 | 0.319 |
| VidKV ($p = 0.2$) | 1.75 | 1.66 | 2.92 | 48.25 | 3.38 | 2.83 | 3.61 | 2.61 | 3.21 | 3.13 | **3.12** | 47.87 | 0.312 |
| Qwen2.5-VL-7B | | | | | | | | | | | | | |
| Baseline | 16-Bit | | 2.93 | 56.53 | 3.20 | 2.91 | 3.36 | 2.71 | 3.31 | 3.10 | 2.95 | 44.23 | 0.334 |
| KIVI | 2-Bit (K-ℂ V-𝕋) | | 2.93 | **55.63** | 3.30 | **2.97** | 3.54 | 2.71 | 3.32 | 3.17 | 2.85 | 43.28 | 0.330 |
| VidKV | 2-Bit (K-ℂ V-ℂ) | | **2.94** | 55.39 | 3.31 | 2.91 | **3.57** | **2.74** | 3.38 | **3.18** | 2.92 | **45.01** | **0.332** |
| VidKV | 1.50 | 2.00 | 2.88 | 54.24 | 3.31 | 2.90 | 3.56 | 2.67 | 3.35 | 3.15 | **2.92** | 44.56 | 0.311 |
| | 1.25 | 2.00 | 2.54 | 50.49 | 3.20 | 2.76 | 3.43 | 2.56 | 3.10 | 3.01 | 2.90 | 44.93 | 0.286 |
| | 2.00 | 1.58 | 3.01 | 52.03 | 3.15 | 2.81 | 3.46 | 2.58 | 3.16 | 3.03 | **2.92** | 42.99 | 0.309 |
| | 1.50 | 1.58 | 2.68 | 49.10 | 3.08 | 2.78 | 3.41 | 2.51 | 3.12 | 3.00 | 2.91 | 43.36 | 0.310 |
| VidKV ($p = 0.2$) | 1.50 | 1.66 | 2.87 | 49.20 | 3.15 | 2.85 | 3.49 | 2.63 | **3.38** | 3.10 | **2.92** | 44.17 | 0.321 |

an analysis of KV-cache quantization in video LLMs; consequently, most baseline methods do not support sub-2-bit (1.x-bit) implementations. In the analyses, the key cache is tested within a quantization range of 1.25 to 2 bits, while the value cache is evaluated with 1.58-bit and 1.66-bit (20% tokens for 2-bit and 80% tokens for 1.58-bit) quantization, both employing *per-channel* quantization.

We evaluate VidKV across multiple video-to-text benchmarks and the video caption benchmark (see Sec. B in the appendix). Results in Tab. 2 show that the models after 2-bit KV cache quantization can achieve comparable or slightly better performance *vs.* their FP16 counterparts. As aforementioned, this motivated us to explore lower-bit quantization at the beginning. Additionally, VidKV is compared with KIVI (Liu et al., 2024c) under 2-bit quantization. Notably, the key distinction between VidKV and KIVI lies in the application of *per-channel* quantization in the value cache. Results indicate that VidKV *outperforms* KIVI across multiple benchmarks, validating the necessity of our KV cache distribution analyses for video LLMs.

For *1.x-bit* quantization, when the key cache is quantized to 1.5 bits, accuracy remains nearly unchanged, demonstrating the effectiveness of our proposed mixed-precision quantization and the FFT-based 1-bit quantization strategy for the key cache. For the LLaVA-OV model, reducing the KV cache precision from 16 bits to 1.5 bits (or even 1.25 bits) and 1.58 bits results in only a minimal accuracy degradation. The Qwen2.5-VL model employs a highly compressed vision token representation relative to other video LLMs. Consequently, a slight degradation in accuracy is observed in the Qwen2.5-VL model—particularly in the TempCompass and VideoDC benchmarks—although the performance remains within an acceptable range. Furthermore, enabling semantic token protection (STP) for the value cache increased the average quantized bit from 1.58 to 1.66, resulting in improved accuracy across multiple benchmarks (marked in yellow), with a notable improvement on VideoChat-GPT, as shown in Tab. 2. Spending less than 0.1 bit in both models allows them to attain accuracy comparable to that of the FP16 configuration.

## 5.3 Ablation Study

**Lower-Bit Key Cache Quantization.** As shown in Fig. 6 (a), this study further investigates the principles and potential of lower-bit quantization for the key cache, building upon the findings of the previous section. While the value cache maintains 2-bit and 1.58-bit quantization, the key cache can be quantized from 1.75-bit to 1.2-bit with only a minor reduction in accuracy. However, a significant performance loss is observed

Table 3: Results of the ablation study of our method in the LLaVA-OV model (see results of Qwen2.5-VL in Sec. D.2). In each pair of comparison results, the superior result is shown in **bold**. STP employs the proposed semantic-based token filtering protection strategy, while RTP protects randomly screened tokens. FFT is exclusively applied alongside 1-bit quantization within mixed-precision quantization.

| | Settings | | | | VideoDC | MovieChat | | TempCompass | VideoChat-GPT | | | | | |
| Bit | FFT | STP | RTP | p | GPT Sco. | GPT Sco. | Acc. | Avgerage | CI | DO | CU | TU | CO | Avg. |
|---|---|---|---|---|---|---|---|---|---|---|---|---|---|---|
| 16-Bit | - | - | - | - | 3.01 | 3.09 | 47.87 | 49.05 | 3.47 | 2.97 | 3.71 | 2.74 | 3.49 | 3.27 |
| K-1.5 / V - 2 | ✗ | ✗ | ✗ | 0.0 | 2.92 | 3.06 | 47.49 | 48.98 | 3.47 | 2.87 | 3.60 | 2.67 | 3.33 | 3.18 |
| K-1.5 / V - 2 | ✓ | ✗ | ✗ | 0.0 | **2.95** | **3.12** | **48.28** | **50.45** | **3.49** | **2.94** | **3.63** | **2.70** | **3.38** | **3.23** |
| K-1.5 / V - 1.66 | ✓ | ✗ | ✓ | 0.2 | 2.89 | 3.11 | 47.01 | 46.36 | 3.26 | 2.77 | 3.54 | 2.63 | 3.10 | 3.06 |
| K-1.5 / V - 1.66 | ✓ | ✓ | ✗ | 0.2 | 2.89 | 3.11 | **47.25** | **47.55** | **3.35** | **2.79** | **3.65** | **2.66** | **3.11** | **3.12** |

when the key cache is quantized below 1.2 bits. These observations indicate that certain abnormal channels in the key cache induce significant quantization errors when subject to 1-bit, implying that effective 1-bit quantization for the key cache remains challenging.

**STP of Value Cache.** Preserving a subset of tokens at higher precision consistently improves final accuracy. Thus, STP is compared against the random selection of an equivalent proportion of tokens, as demonstrated in Tab. 5, where STP outperforms random selection. Additionally, as shown in Fig. 6 (b), we investigate the trade-off between precision and model performance using the STP method. Our results indicate that as $p$ increases, the average bit number of the value cache initially improves, resulting in a gradual enhancement of model performance. However, 1-bit quantization of the value cache results in unacceptable performance degradation.

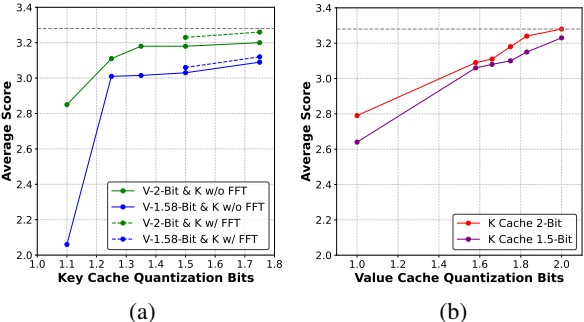

(a)          (b)

Figure 6: Key cache shows a sharp performance drop at 1.1-bit quantization. For value cache, the average bit width rises from 1.58 to 2 as $p$ increases from 0 to 1.

**FFT-based 1-Bit Quantization.** Tab. 5 (marked in green) shows the performance differences between using and not using the proposed FFT-based 1-bit quantization within the mixed-precision quantization for the key cache. The results indicate that the application of FFT enhances model performance across all benchmarks. Furthermore, as shown in Tab. 4 (marked in green), applying FFT to the video caption task leads to a significant improvement in model quantization accuracy. Combined with FFT, a similar trend is also observed in Fig. 6 (a). This proves that the quantization of the cache after transforming it into the frequency domain by FFT is reasonable and effective.

## 6 CONCLUSION

This paper presents **VidKV**, the *first* KV cache quantization method for video LLMs. At its core, VidKV employs a mixed-precision strategy to quantize key and value caches separately with specialized schemes: (1) key cache is quantized to 2 bits and 1 bit, where a novel FFT-based quantization scheme is introduced for the 1-bit quantization, which effectively mitigates the performance drop; (2) value cache is quantized to 2-bits and 1.58 bits (+1/-1/0), where we importantly find the value cache should also be quantized in a *per-channel* fashion, instead of the *per-token* fashion as argued by prior counterpart methods for LLMs (KIVI), implying KV cache quantization for *video LLMs* is different from that for *LLMs*. Extensive experiments on six standard benchmarks show that we achieve 1.5-bit and 1.58-bit KV cache quantization without significant performance loss. Notably, the method is training-free and plug-and-play.

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

## A DETAILED IMPLEMENTATIONS

### A.1 ALGORITHM

In this section, we present the algorithm for VidKV as discussed in Sec. 4 (algorithms 1 to 3). algorithm 1 details the computation process of VidKV during both the prefilling and decoding phases, while algorithms 2 and 3 respectively present the custom functions employed.

---

**Algorithm 1:** Algorithm of VidKV

---

**parameter:** Group size $G$, residual length $R$, hyperparameters $p, k$

**procedure** Prefill**:**

    **Input:** $\boldsymbol{X} \in \mathbb{R}^{l \times d}$

    $\boldsymbol{X}_K = \boldsymbol{X}\boldsymbol{W}_K, \boldsymbol{X}_V = \boldsymbol{X}\boldsymbol{W}_V$

    $\boldsymbol{r} = l \% G$

    $\boldsymbol{X}_{V_q} = \boldsymbol{X}_V[: l - \boldsymbol{r}], \boldsymbol{X}_{V_r} = \boldsymbol{X}_V[l - \boldsymbol{r} :]$

    $\boldsymbol{X}_{K_q} = \boldsymbol{X}_K[: l - \boldsymbol{r}], \boldsymbol{X}_{K_r} = \boldsymbol{X}_K[l - \boldsymbol{r} :]$

    **if** $p > 0$ **then**

        $Q(\boldsymbol{X}_{V_q}) \leftarrow \text{STPQuant}(\boldsymbol{X}_{V_q})$

    **end**

    **else**

        $Q(\boldsymbol{X}_{V_q}) \leftarrow \text{1.58Quant}(\boldsymbol{X}_{V_q})$

    **end**

    $Q(\boldsymbol{X}_{K_g}) \leftarrow \text{MixQuant}(\boldsymbol{X}_{K_q}, k)$

    KV cache $\leftarrow Q(\boldsymbol{X}_{K_q}), \boldsymbol{X}_{K_r}, Q(\boldsymbol{X}_{V_q}), \boldsymbol{X}_{V_r}$

    **return** $\boldsymbol{X}_K, \boldsymbol{X}_V$

**end**

**procedure** Decoding**:**

    **Input:** KV cache, $\boldsymbol{x} \in \mathbb{R}^{1 \times d}$

    $Q_x = \boldsymbol{x}\boldsymbol{W}_Q, K_x = \boldsymbol{x}\boldsymbol{W}_K, V_x = \boldsymbol{x}\boldsymbol{W}_V$

    $Q(\boldsymbol{X}_{K_q}), \boldsymbol{X}_{K_r}, Q(\boldsymbol{X}_{V_q}), \boldsymbol{X}_{V_r} \leftarrow$ KV cache

    $\boldsymbol{X}_{K_r} \leftarrow \text{Concat}([\boldsymbol{X}_{K_r}, \boldsymbol{x}_K], \text{dim=token})$

    $\boldsymbol{X}_{V_r} \leftarrow \text{Concat}([\boldsymbol{X}_{V_r}, \boldsymbol{x}_V], \text{dim=token})$

    $\boldsymbol{X}'_{K_q} \leftarrow \text{DeQuant}(Q(\boldsymbol{X}_{K_q}))$

    $\boldsymbol{X}_K \leftarrow \text{Concat}([\boldsymbol{X}'_{K_q}, \boldsymbol{X}_{K_r}], \text{dim=token})$

    $\boldsymbol{A}_w \leftarrow \text{Softmax}(\boldsymbol{x}_Q \boldsymbol{X}_K^\top), \text{dim=token}$

    $\boldsymbol{x}_O \leftarrow \boldsymbol{A}_w Q(\boldsymbol{X}_{V_q}) + \boldsymbol{A}_w \boldsymbol{X}_{V_r}$

    **if** $len(\boldsymbol{X}_{V_r}) = R$ **then**

        $Q(\boldsymbol{X}_{V_r}) \leftarrow \text{1.58Quant}(\boldsymbol{X}_{V_r})$

        $Q(\boldsymbol{X}_{V_q}) \leftarrow \text{Concat}([Q(\boldsymbol{X}_{V_q}), Q(\boldsymbol{X}_{V_r})])$

        $\boldsymbol{X}_{V_r} \leftarrow$ empty tensor.

        $Q(\boldsymbol{X}_{K_q}) \leftarrow \text{MixQuant}(\boldsymbol{X}_K)$

        $\boldsymbol{X}_{K_r} \leftarrow$ empty tensor.

    **end**

    KV cache $\leftarrow Q(\boldsymbol{X}_{K_q}), \boldsymbol{X}_{K_r}, Q(\boldsymbol{X}_{V_q}), \boldsymbol{X}_{V_r}$

    **return** $\boldsymbol{x}_O$

**end**

---

---

**Algorithm 2:** Function of 1.58-Bit Quantization

---

**parameter:** Group size $G$, residual length $R$, important token index mask $E$, hyperparameters $p, k, \gamma$

**function** `1.58Quant`$(\boldsymbol{X}_{V_q})$**:**

$\quad \boldsymbol{s} \leftarrow \text{Mean}(|\boldsymbol{X}_{V_q}|, \text{dim=channel})$

$\quad \alpha \leftarrow \gamma \boldsymbol{s}$

$\quad Q(\boldsymbol{X}_{V_q}) = \begin{cases} 1, & \boldsymbol{x}_v > \alpha, \\ -1, & \boldsymbol{x}_v < -\alpha, \\ 0, & \text{otherwise} \end{cases}$

$\quad$ **return** $Q(\boldsymbol{X}_{V_q})$

**end**

**function** `STPQuant`$(\boldsymbol{X}_{V_q})$**:**

$\quad \boldsymbol{X}_{V_q^1} \leftarrow \boldsymbol{X}_{V_q}[E]$

$\quad \boldsymbol{X}_{V_q^2} \leftarrow \boldsymbol{X}_{V_q}[\sim E]$

$\quad Q(\boldsymbol{X}_{V_q^1}) \leftarrow \text{GQuant}(\boldsymbol{X}_{V_q^1}, \text{d=channel}, \text{bit=2})$

$\quad Q(\boldsymbol{X}_{V_q^2}) \leftarrow $ `1.58Quant`$(\boldsymbol{X}_{V_q^2})$

$\quad$ **return** $[Q(\boldsymbol{X}_{V_q^1}), Q(\boldsymbol{X}_{V_q^2})]$

**end**

---

---

**Algorithm 3:** Function of Key Mix-Quantization

---

**parameter:** Group size $G$, residual length $R$

**function** `MixQuant`$(\boldsymbol{X}_{K_q}, k)$**:**

$\quad \boldsymbol{x}_k \leftarrow \text{flatten}(\boldsymbol{X}_{K_q})$

$\quad$ **range** $\leftarrow \text{Max}(\boldsymbol{x}_k) - \text{Min}(\boldsymbol{x}_k)$

$\quad$ **D-mask** $\leftarrow \text{TopK}(\textbf{range}, k)$

$\quad$ **N-mask** $\leftarrow \sim$ **D-mask**

$\quad \boldsymbol{X}_{\text{nom}} \leftarrow \boldsymbol{X}_{K_q}[\textbf{N-mask}]$

$\quad Q(\boldsymbol{X}_{K_q^1}) \leftarrow$ `1BitQuant`$(\boldsymbol{X}_{\text{nom}}, k)$

$\quad \boldsymbol{X}_{\text{abn}} \leftarrow \boldsymbol{X}_{K_q}[\textbf{D-mask}]$

$\quad Q(\boldsymbol{X}_{K_q^2}) \leftarrow \text{GQuant}(\boldsymbol{X}_{\text{abn}}, \text{d=channel}, \text{bit=2})$

$\quad$ **return** $[Q(\boldsymbol{X}_{K_q^1}), Q(\boldsymbol{X}_{K_q^2})]$

**end**

**function** `1BitQuant`$(\boldsymbol{X}, k)$**:**

$\quad$ **if** $k$ *in [0.5, 0.75]* **then**

$\quad\quad \boldsymbol{X}_{\text{fft}} \leftarrow \text{FFT}(\boldsymbol{X})$

$\quad\quad \boldsymbol{X} \leftarrow \text{Reshape}(\boldsymbol{X}_{\text{fft}}, [l, d \times 2])$

$\quad$ **end**

$\quad Q(\boldsymbol{X}_q) \leftarrow \text{GQuant}(\boldsymbol{X}, \text{d=channel}, \text{bit=1})$

$\quad$ **return** $Q(\boldsymbol{X}_q)$

**end**

---

## A.2 TASKS

VideoDC is a benchmark for single-video description. VideoChat-GPT comprises five subtasks: CI stands for correctness of information, DO stands for detail orientation, CU stands for contextual understanding, TU

stands for temporal understanding, and CO stands for consistency. These metrics are assessed using an LLM-generated prediction score ranging from 0 to 5 (GPT Score). MovieChat assesses a model's comprehension ability to long videos, evaluated through a combination of GPT Score and accuracy. TempCompass evaluates five key aspects: action, speed, direction, attribute change, and event order. For KV cache evaluation, we use the caption branch task on TempCompass and only test the text generation task on WorldQA. Finally, VATEX is a specialized video caption generation benchmark, and its accuracy is assessed using four metrics: BLEU Papineni et al. (2002), METEOR (Denkowski & Lavie, 2014), ROUGE-L (Lin, 2004), and CIDEr (Vedantam et al., 2015).

## B   RESULTS ON VIDEO CAPTION BENCHMARK

For the video captioning task, VATEX is used for evaluation. As shown in Tab. 4, once again, no accuracy loss is observed for the quantization of the 2-bit KV cache, and we can get better results when using per-channel quantization for the value cache. However, for *1.x-bit* quantization (K-1.5, V-1.58), a decline is observed across the four evaluation metrics, though some accuracy recovery is achieved through STP. Considering the distinction between VATEX and other datasets that utilize GPT-based scoring, the four evaluation metrics used by VATEX are hard indicators, which are more sensitive to variations in the generated text and exhibit lower flexibility. Under this strict evaluation environment, our proposed VidKV continues to demonstrate acceptable performance.

Table 4: Comparison of different quantization settings on VATEX benchmarks.

| Method | Settings | | | VATEX | | | |
|---|---|---|---|---|---|---|---|
| | K-(Bit) | V-(Bit) | FFT | BLEU-4 | Meteor | Rouge-L | CIDEr |
| LLaVA-OV-7B | | | | | | | |
| Baseline | 16-Bit | | - | 14.88 | 19.85 | 39.25 | 27.42 |
| KIVI | 2-Bit (K-ℂ V-𝕋) | | - | 14.33 | 19.55 | 38.67 | 26.27 |
| VidKV | 2-Bit (K-ℂ V-ℂ) | | ✗ | 15.24 | 19.79 | 39.45 | 27.57 |
| VidKV | 1.5-Bit | 2-Bit | ✗ | 14.06 | 18.91 | 38.11 | 23.38 |
| | 1.5-Bit | 2-Bit | ✓ | 14.96 | 19.47 | 39.01 | 25.91 |
| | 1.5-Bit | 1.58-Bit | ✓ | 14.06 | 16.43 | 35.28 | 20.09 |
| VidKV ($p = 0.2$) | 1.5-Bit | 1.66-Bit | ✓ | 15.31 | 17.99 | 37.24 | 23.06 |
| Qwen2.5-VL-7B | | | | | | | |
| Baseline | 16-Bit | | - | 19.17 | 20.43 | 40.99 | 41.87 |
| KIVI | 2-Bit (K-ℂ V-𝕋) | | - | 19.20 | 21.26 | 41.66 | 41.98 |
| VidKV | 2-Bit (K-ℂ V-ℂ) | | ✗ | 19.97 | 21.26 | 42.06 | 43.15 |
| VidKV | 1.5-Bit | 2-Bit | ✓ | 19.46 | 21.02 | 42.04 | 41.86 |
| | 2-Bit | 1.58-Bit | ✗ | 13.61 | 17.62 | 36.90 | 28.86 |
| | 1.5-Bit | 1.58-Bit | ✓ | 13.09 | 17.44 | 35.88 | 29.76 |
| VidKV ($p = 0.2$) | 1.5-Bit | 1.66-Bit | ✓ | 14.63 | 17.99 | 37.75 | 31.72 |

## C   MORE OBSERVATIONS AND FUTURE WORK

In Sec. 1 and Sec. 4.1, we analyzed the distribution characteristics of the KV cache in VideoLLMs. However, the distinct temporal characteristics of video data warrant further analysis. As illustrated in Fig. 1, the distribution of the KV cache across each channel in VideoLLMs exhibits regularity and periodicity—particularly within the value cache—which contrasts with findings from previous studies on models such as Llama (Liu et al., 2024c; Hooper et al., 2024). We attribute this phenomenon to the reliance of most current VideoLLMs on the sequential concatenation of video tokens. Within tokens corresponding to a video frame, tokens occupying identical positions frequently convey similar information and exhibit uniform distribution patterns, resulting in distinctive regularity that may inform strategies such as token screening or reordering. This observation will represent a major direction for our future research. Additionally, we recognize the high redundancy inherent in visual tokens, and we will focus on strategies such as token pruning and merging in future work.

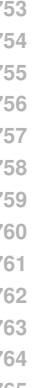

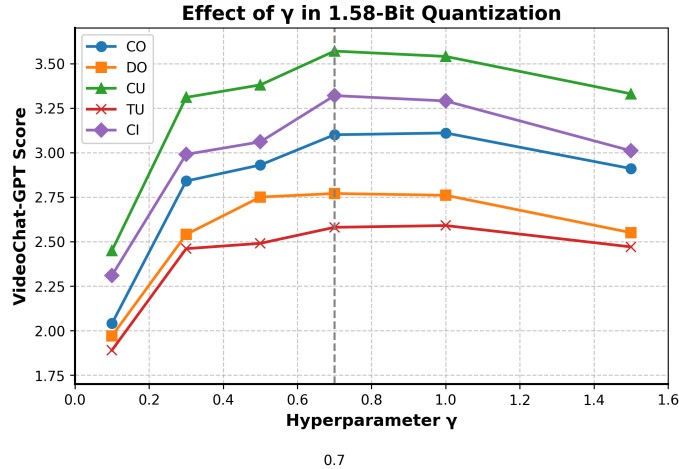

0.7

Figure 7: **Ablation study of** $\gamma$**.** CI stands for correctness of information, DO stands for detail orientation, CU stands for contextual understanding, TU stands for temporal understanding, and CO stands for consistency.

## D    MORE ABLATION STUDY

### D.1    ABLATION STUDY ABOUT THE WEIGHT $\gamma$

Fig. 7 presents a visual comparison of the impact of different $\gamma$ settings on 1.58-bit quantization accuracy. The results indicate that, contrary to conventional assumptions, the optimal performance is not attained when $\gamma = 1$. Instead, the highest benchmark test performance is observed when $\gamma = 0.7$. A significantly lower $\gamma$ value adversely impacts model performance, suggesting that the chosen 1.58-bit quantization threshold hyperparameter is both reasonable and effective.

### D.2    FFT-BASED 1-BIT QUANTIZATION FOR QWEN2.5-VL.

Table 5: Results of the ablation study for Qwen2.5-VL. In each pair of comparison results, the superior result is shown in **bold**. FFT is exclusively applied alongside 1-bit quantization within mixed-precision quantization.

| Settings | | | | | VideoDC | MovieChat | | TempCompass | VideoChat-GPT | | | | | |
|---|---|---|---|---|---|---|---|---|---|---|---|---|---|---|
| **Bit** | **FFT** | **STP** | **RTP** | **p** | **GPT Sco.** | **GPT Sco.** | **Acc.** | **Avgerage** | **CI** | **DO** | **CU** | **TU** | **CO** | **Avg.** |
| Qwen2.5-VL-7B | | | | | | | | | | | | | | |
| 16-Bit | - | - | - | - | 2.93 | 2.95 | 44.23 | 56.53 | 3.20 | 2.91 | 3.36 | 2.71 | 3.31 | 3.10 |
| K-1.5 / V - 2 | ✗ | ✗ | ✗ | 0.0 | 2.81 | 2.92 | 44.27 | 52.32 | 3.30 | 2.86 | 3.55 | 2.72 | 3.29 | 3.14 |
| K-1.5 / V - 2 | ✓ | ✗ | ✗ | 0.0 | **2.88** | **2.94** | **44.89** | **54.24** | **3.34** | **2.95** | **3.58** | **2.87** | **3.31** | **3.21** |

## E    STATEMENT OF LLMS

This work used large language models (LLMs) solely to polish language and improve manuscript readability. No LLMs were used for data generation, analysis, or interpretation of results. All scientific details, methodologies, and findings reported here are the authors' original contributions.

## F  FURTHER DISCUSSION ON 1-BIT QUANTIZATION

This study provides an initial investigation into low-bit KV cache quantization (*1.x-bit*) for video LLMs. Empirical results across multiple benchmark programs indicate that maintaining 1.5-bit quantization for the key cache and 1.58-bit quantization for the value cache results in negligible accuracy degradation. Nonetheless, extreme 1-bit quantization remains highly challenging and frequently results in model collapse. Tab. 6 presents a comparative evaluation of the proposed VidKV and KIVI (Liu et al., 2024c) under 1-bit quantization. Although VidKV experiences substantial performance degradation under 1-bit quantization, it still outperforms KIVI significantly. Future research will focus on advancing low-bit KV cache quantization to minimize bit-width while approaching the theoretical lower limit.

| Settings | | VideoDC | MovieChat | | TempCompass | WorldQA |
|---|---|---|---|---|---|---|
| Method | Bit | GPT Sco. | GPT Sco. | Acc. | Avg. | GPT Sco. |
| Baseline | 16-Bit | 3.01 | 3.09 | 47.87 | 49.05 | 0.33 |
| KIVI | 1-Bit | 0.99 | 0.53 | 0.910 | 2.45 | - |
| Ours | 1-Bit | **1.25** | **2.51** | **31.15** | **12.8** | **0.15** |
| Task | | VideoChat-GPT | | | | |
| Method | Bit | CI | DO | CU | TU | CO |
| Baseline | 16-Bit | 3.47 | 2.97 | 3.71 | 2.74 | 3.49 |
| KIVI | 1-Bit | 0.66 | 1.07 | 0.94 | 0.95 | 1.22 |
| Ours | 1-Bit | **1.08** | **1.40** | **1.53** | **1.22** | **1.33** |

Table 6: Results of 1-Bit Quantization for KV Cache. The "-" symbol indicates complete model failure.

## G  DISCUSSION

Unlike previous studies, we introduce two distinct quantization strategies for key and value cache, respectively. Our findings indicate that the distribution characteristics of the two caches differ, making it challenging to directly apply the key cache's mixed-precision quantization strategy to the value cache. Thus, a more efficient and suitable approach, 1.58-bit quantization, is selected for the value cache. This approach retains almost all the advantages of 1-bit quantization and yields strong results. An attempt was also made to apply 1.58-bit quantization to the key cache, but it proved ineffective due to significant variations in the channel dimension. Accordingly, the proposed two different strategies for KV caching are based on their unique distribution characteristics, with extensive experiments confirming their effectiveness.

