# OpenReview forum: "Plug-and-Play 1.x-Bit KV Cache Quantization for Video Large Language Models"
_ICLR.cc/2026/Conference — ICLR 2026 Conference Withdrawn Submission_

### Official Review · Reviewer_fUDc · 2025-10-27

**Soundness:** 2
**Presentation:** 3
**Contribution:** 2
**Rating:** 4
**Confidence:** 4

**Summary:**

This paper presents VidKV, a study on how to effectively quantize the KV cache for video LLMs. The authors propose to quantize the key cache per channel using FFT-based 1-bit quantization on normal channels and apply 2-bit quantization to abnormal channels. They also claim that, contradict to prior works on text-only LLM, the value cache, instead, should be quantized per-channel, and they choose to apply 1.58-bit quantization to the value cache, along with 2-bit quantization for those critical visual tokens. This mixed precision quantization strategy to some extent preserves the quality of the model's output, as shown in the experiment section.

**Strengths:**

1. The motivation of this paper, pushing the quantization level to 1.x bit quantization for video LLMs, is clear, and the overall presentation of the paper is fluent and clean.

2. The authors present several ablation studies to justify their design choices for how to quantize key/value cache in a better way, which is informative and well supportive of their methodology.

**Weaknesses:**

1. The claim that the value cache for video LLMs should be quantized per-channel instead of per-token is not well justified. The primary justification for this claim is presented in Table 1. However, in Table 1, when the Bit (K/V) is 1.5/1.58, it is shown that using per-token quantization for the value cache is better. Conversely, when the Bit (K/V) is 2/2, per-channel is better. The results presented here are very confusing.

2. When it comes to the experiment section, I doubt the effectiveness of the proposed methodology, as there are essentially no baseline results for reference when the quantization bits are below 2 bits, and it is very hard to assess and interpret the evaluation results. For example, one naive baseline can be applying 1.58bits to both key/value cache as presented in the original paper for BitNet b1.58 [1]. Also, it is helpful to compare the proposed methodology to existing KV cache quantization works for VLMs, as these models also accept vision input and should potentially be applicable to video input.

3. The authors claim that "Notably, KV cache quantization in VideoLLMs is essential to mitigate memory and computational bottlenecks." around line 105. However, there is no computation comparison presented in this paper. Also, I am curious if the proposed methodology indeed improved the efficiency of video LLM inference, considering the complex quantization scheme the authors used in this paper. Any latency comparison results would be necessary to clarify this.

4. This paper focuses heavily on the methodology and empirical results, but does not present adequate insights into how LLMs with vision input differ from LLMs with only text input. For example, does the vision token behave similarly from the quantization perspective compared to text tokens? The author claims that the key cache should be quantized per channel, but why? Is it because of the specialty of vision tokens? The novelty and contribution of the proposed methodology appear limited, as it primarily combines existing techniques. Nonetheless, deriving new and well-justified insights (if there are any) by applying established methods to new domains could still offer meaningful value to the community. I would be happy to see more discussion on insights in the main paper.

[1] The Era of 1-bit LLMs: All Large Language Models are in 1.58 Bits

**Questions:**

1. The results presented in Table 1 are inconsistent, as K-C, V-C is better when Bit (K/V) is 2 / 2, but worse when Bit (K/V) is 1.5 / 1.58. Why?

2. In Figure 6, the performance drops significantly without FFT as the quantization bit approaches 1-bit. However, the results with FFT are not provided. How does the performance behave when FFT is applied under such extreme quantization (near 1-bit)?

---

> ### Author Response · Authors · 2025-11-20
> **Response to Reviewer fUDc**
>
> We thank the reviewer for recognizing our work. We responded to the reviewer's questions in detail:
>
> > **Q1**: The results presented in Table 1 are very confusing.
>
> **A1**: Thanks for the feedback. Table 1 contains a marking error, which we have corrected in the new version (highlighted in blue). In all subsequent experiments, VidKV adopts the proposed configuration in which both the K-cache and V-cache are quantized on a per-channel basis. The updated Table 1 indicates that per-channel quantization achieves superior results for both 2-bit and 1.x-bit settings. Moreover, the performance gap is more pronounced under 1.x-bit quantization. We sincerely apologize for any inconvenience our previous submission may have caused.
>
> > **Q2**: Compare with BitNet b1.58.
>
> **A2**: We thank the reviewer for the insightful comments. Following your suggestion, we conducted a comparative experiment with BitNet b1.58. As shown in the table below, we compared applying BitNet b1.58 to the key and value caches. The results show that VidKV achieves superior performance owing to a more effective 1.58-bit quantization scheme and the explicit handling of abnormal channels in the key cache.
> | Method        | Avg. Bit | CI   | DO   | CU   | TU   | CO   | Avg. |
> |--------------|---------:|:----:|:----:|:----:|:----:|:----:|:----:|
> | Baseline     | FP16 | 3.47 | 2.97 | 3.71 | 2.74 | 3.49 | 3.27 |
> | VidKV        | 1.54 | 3.32 | 2.77 | 3.57 | 2.58 | 3.10 | 3.06 |
> | BitNet b1.58 | 1.58 | 0.23 | 0.25 | 0.37 | 0.15 | 0.17 | 0.39 |
>
>
> > **Q3**: The problem of inference efficiency under mixed-precision quantization.
>
> **A3**: Thank you for your question. Given the complexity of mixed-precision quantization, we are also actively developing a more efficient kernel implementation. Currently, in the absence of such kernel support, we must dequantize back to FP16 for the attention computation, which makes it difficult to provide reliable measurements of inference efficiency. It is worth noting that common mixed-precision quantization algorithms, such as SKVQ [R1], rely on fake quantization to evaluate algorithm performance. In contrast, VidKV preserves Int4 representations when storing the KV cache, thereby directly reducing the KV cache size compared to fake quantization. In the ideal case, this design can reduce the KV-cache size by more than 80%. Consequently, more inference batches can be processed under the same computational budget, and the end-to-end prediction throughput can, in principle, achieve a 7–8 $\times$ speedup.
>
> > **Q4**: Insights about how LLMs with vision input differ from LLMs with only text input.
>
> **A4**: Thank you for your valuable suggestions. The starting point of this work is the observation that, in VideoLLMs, the redundancy of video tokens causes standard 2-bit KV-cache quantization to have only a minor impact on model accuracy, which in turn motivates the exploration of lower-bit quantization schemes. By contrast, for text-only LLMs, the focus remains on 2-bit or 4-bit quantization schemes that are effectively lossless. Second, the adoption of per-channel quantization is motivated by the characteristics of the value cache in VideoLLMs and is empirically validated by the experiments in this work. Finally, we combine the token-pruning algorithm with our 2-bit quantization scheme. As shown in the table below, pruning redundant video tokens causes the accuracy of 2-bit quantization for VideoLLMs to degrade progressively, which further supports the need for specialized KV-cache quantization algorithms for VideoLLMs.
> | VideoChat-GPT Benmark | 100% Tokens |75% Tokens | 50% Tokens | 25% Tokens |
> |--------|--------|--------|--------|--------|
> | 2-bit | 3.27 | 3.25 | 3.17 | 2.95 |
> | FP16 | 3.27 | 3.27 | 3.24 | 3.20 |
>
> Our experiments are conducted on LLaVA-OV-7B with VisionZip [R2] chosen as the token compression algorithm.
>
>
> > **Q5**: In Figure 6, the performance drops significantly without FFT as the quantization bit approaches 1-bit. However, the results with FFT are not provided. How does the performance behave when FFT is applied under such extreme quantization (near 1-bit)?
>
> **A5**: Thank you for your valuable suggestions. As discussed in Appendix F, Table 6 shows that 1-bit quantization using FFT outperforms the baseline without FFT (KIVI) by a substantial margin. These results demonstrate the effectiveness of the proposed method.
>
> [R1] Duanmu H, Yuan Z, Li X, et al. SKVQ: Sliding-window Key and Value Cache Quantization for Large Language Models, COLM 2024.
>
> [R2] Yang S, Chen Y, Tian Z, et al. Visionzip: Longer is better but not necessary in vision language models, CVPR 2025.

---

### Official Review · Reviewer_Jkph · 2025-10-28

**Soundness:** 4
**Presentation:** 4
**Contribution:** 2
**Rating:** 4
**Confidence:** 5

**Summary:**

This paper proposes VidKV, a PTQ framework for Video LLM KV-Cache compression. VidKV combines 2-bit quantization with FFT-based 1-bit quantization for Key-Cache; for the value cache, it uses 1.58-bit quantization with selective token protection. A key finding of the paper is that, unlike plain text LLMs, the value cache of VideoLLMs is more amenable to per-channel rather than per-token quantization. Experiments demonstrate that VidKV can compress the KV cache to approximately 1.5 bits with little performance degradation compared to the FP16 version.

**Strengths:**

1. Pioneering Work in Sub-2-Bit Quantization for Video LLM: The paper fills an important gap by exploring sub-2-bit KV cache quantization specifically for VideoLLMs, a domain that has been underexplored compared to textLLMs.
2. Strong Performance at Near-2-Bit Precision: The experimental results compellingly show that the proposed method achieves nearly lossless performance compared to the FP16 baseline when quantizing the KV cache to approximately 2-bits.

**Weaknesses:**

1. Lack of Essential Efficiency Experiments: The paper's claims are centered on improving inference efficiency, yet it completely lacks empirical data on crucial metrics like end-to-end throughput or latency. All evaluations are focused on model accuracy, which is insufficient for a work positioned to solve an efficiency bottleneck. This is a critical omission.
2. Incremental Contribution: The primary novel contribution appears to be the observation that the value cache benefits from per-channel quantization. However, the experimental results in Table 2 show that applying 2-bit per-channel quantization to the value cache (VidKV 2-bit) does not offer a significant performance improvement over the KIVI baseline, which uses per-token quantization. Other techniques build heavily upon concepts thoroughly explored in prior work like KIVI. This makes the overall contribution to the work appear incremental.

**Questions:**

1. FFT Overhead on GPUs: The paper claims that the FFT overhead is small ("less than 5%"). Could you provide specific latency figures for the FFT/IFFT operations on a GPU? How does this overhead scale with an increasing batch size during the decoding stage?
2. System Complexity of Mixed-Precision: The introduction of mixed-precision quantization (e.g., 1-bit, 1.58-bit, 2-bit) and token protection adds complexity to KV cache management. How would this affect integration into highly optimized inference systems like vLLM? Does this introduce significant system-level complexity that could offset theoretical memory savings?
3. Clarification on Figure 3: The legend in Figure 3 appears to be confusing, with multiple legend items sharing the same visual representation (e.g., "Important Token" and "Important Token in V"). Could you please clarify this visualization?

---

> ### Author Response · Authors · 2025-11-22
> **Response to Reviewer Jkph**
>
> We thank the reviewer for recognizing our work. We responded to the reviewer's questions in detail:
>
> > **Q1**: Lack of Essential Efficiency Experiments.
>
> **A1:** We thank the reviewer for raising this point. We agree that end-to-end efficiency is crucial for methods targeting the KV cache bottleneck. However, our current implementation is a prototype and does not yet include a kernel that operates directly on the Int4 KV representation. In practice, we store the KV cache in Int4 format but dequantize it back to FP16 before each attention computation. In the revision, we will explicitly clarify this limitation and provide a more detailed discussion of efficiency. Unlike common mixed-precision schemes such as SKVQ [R1], which rely on fake quantization and keep the KV cache in FP16 during inference, VidKV instead stores the KV cache in a true Int4 format, thereby reducing its size by more than 80% in long-context generation. This design directly alleviates memory-capacity and bandwidth bottlenecks and, when combined with a dedicated Int4 attention kernel (which we are actively developing), can, in principle, translate into a 7–8× throughput improvement in KV-bound deployment scenarios.
>
> > **Q2**: Incremental Contribution.
>
> **A2**: We thank the reviewer for the insightful comments. Our motivation stems from the increased error tolerance of KV-cache quantization in VideoLLMs, where the redundancy of video tokens makes it worthwhile to explore lower-bit quantization schemes. Besides, the reason why per-channel finding is less evident at 2-bit precision is that 2-bit quantization is nearly lossless for redundant tokens. However, Table 1 demonstrates that, under 1.58-bit quantization, per-channel quantization is the optimal choice for the value cache.
>
> > **Q3**: FFT Overhead on GPUs.
>
> **A3**: We thank the reviewer for the insightful comments. We empirically evaluated the computational overhead of the IFFT and FFT operations. The assessment on a single A6000 GPU shows that the time for one IFFT and one FFT is less than $10^{-4}$ seconds (≈0.04 ms), which has a negligible impact on the inference process (a typical decoding step takes about 30 ms).
>
> > **Q4**: System Complexity of Mixed-Precision.
>
> **A4**: Thank you for your valuable suggestions. We acknowledge that mixed-precision quantization increases overall system complexity, but we believe it can still be effectively integrated into modern inference frameworks. This work demonstrates the superiority of the proposed VidKV method at the algorithmic level. We anticipate that, with appropriately optimized kernel implementations, the additional complexity introduced by mixed-precision quantization can be largely mitigated in future deployments. We will include this discussion in the new version.
>
> > **Q5**: Clarification on Figure 3
>
> **A5**: We sincerely apologize for any inconvenience our previous submission may have caused. “Important Token” and “Important Token in V” both denote tokens identified as important based on cross-modal similarity scores. “Important Token in V” specifically denotes the value-cache vector corresponding to an important token in the value cache. We will clarify this terminology more explicitly in the revised manuscript.
>
> [R1] Duanmu H, Yuan Z, Li X, et al. SKVQ: Sliding-window Key and Value Cache Quantization for Large Language Models, COLM 2024.

---

> ### Comment · Reviewer_Jkph · 2025-11-24
>
> 1. "SKVQ relies on fake quantization and keeps the KV cache in FP16 during inference." Using fake quantization to evaluate accuracy is a common and acceptable practice. However, accuracy evaluation and efficiency evaluation are fundamentally different tasks. Implementing int4 storage with dequantization to BF16/FP16 before attention requires only a few lines of torch code; this does not prove anything substantial, nor does it demonstrate that vidkv makes a more meaningful contribution than prior methods such as SKVQ or KVQuant.
>
> 2. According to the authors’ reply, one IFFT and one FFT together take around 0.04 ms. But this operation is performed in every layer, correct? If so, the accumulated overhead is non-negligible. Is there any quantitative analysis of the end-to-end latency overhead? Please report how this overhead scales with batch size, model size, and the volume of the KV cache.
>
> 3. Integrating mixed-precision techniques into systems like vLLM and SGLang is highly complex; it is not a problem that a single kernel can solve. While this may be beyond the scope of the paper and I will not downgrade the submission for this reason, but the authors should be aware of the significant system-level challenges involved.

---

### Official Review · Reviewer_uDCi · 2025-11-01

**Soundness:** 2
**Presentation:** 3
**Contribution:** 3
**Rating:** 4
**Confidence:** 4

**Summary:**

This paper introduces VidKV, a KV cache quantization method for Video Large Language Models (VideoLLMs). VidKV can compress the key-value cache to 1.58-bit precision with minimal performance loss. The authors observe that VideoLLMs exhibit high visual token redundancy, enabling aggressive quantization beyond the typical 2-bit approaches used for text-only LLMs. VidKV employs distinct mixed-precision quantization strategies for keys and values: (1) for key cache, it identifies anomalous channels using range-based evaluation and applies 2-bit quantization to these channels while using FFT-based 1-bit quantization for normal channels; (2) for value cache, it implements 1.58-bit ternary quantization (mapping to {-1, 0, 1}) with an optional semantic token protection mechanism that preserves critical visual tokens at 2-bit precision. Critically, the authors find that value cache in VideoLLMs should use per-channel quantization rather than the per-token approach proposed for text LLMs. Experiments on LLaVA-OV-7B and Qwen2.5-VL-7B across six benchmarks demonstrate that VidKV reduces KV cache size by 80% while maintaining near-baseline performance, offering significant memory savings for long video inference scenarios.

**Strengths:**

1. The paper studies <2-bit KV cache quantization specifically for VideoLLMs. The authors provide a distribution analysis showing that VideoLLMs exhibit distinct characteristics compared to text-only LLMs, particularly the high redundancy of visual tokens and different outlier patterns in value caches.

2. The author applies FFT to the key cache for stabilized low-bit quantization. As observed, the key cache contains channels with sharp fluctuations (or outliers) and abnormal variations in the time domain, and this makes 1-bit quantization extremely challenging and leads to large quantization errors. To handle this, they utilize the FFT transform to the key cache from the time domain to the frequency domain.

3. They propose a mixed-precision quantization scheme that applies different bit-widths to different components based on their distribution characteristics: for the key, they use a range-based channel evaluation to identify outlier channels with large magnitude variations and quantize them at 2-bit precision, while normal channels undergo FFT-based 1-bit quantization in the frequency domain to stabilize their distributions; for the value cache, they employ 1.58-bit ternary quantization ({-1, 0, 1}) across all channels using per-channel quantization, with an optional semantic token protection mechanism that selectively preserves critical visual tokens at 2-bit precision to better trade off between compression ratio and model performance.


4. The method achieves promising results on two VideoLLM families (LLaVA-OneVision-7B and Qwen2.5-VL-7B) across six diverse benchmarks, compressing KV cache to 1.5-bit and 1.58-bit precision with 80% memory reduction and minimal accuracy loss compared to FP16 baselines, while outperforming KIVI at 2-bit quantization through per-channel value cache quantization, demonstrating the practical effectiveness of their training-free, plug-and-play approach.

**Weaknesses:**

1. The outlier in the key cache is already widely observed and studied through very different approaches. For example, KIVI also addresses this through column-wise (per-channel) quantization on the key cache, and KVQuant also observed this phenomenon. The proposed approach with FFT is not only applicable to VideoLLMs but is a general technique for smoothing distributions that has been widely used for outlier handling in various quantization contexts (as acknowledged by citing Tseng et al., 2024). The novelty of applying FFT to key cache quantization is somewhat incremental, and the paper lacks sufficient comparison with other outlier handling techniques such as rotation-based methods (e.g., QuaRot, RotateKV mentioned in related work) or more sophisticated mixed-precision strategies that could potentially achieve similar or better results without the overhead of FFT/IFFT transformations.

2. The benchmark methods seem not sufficient as there are various approaches on KV cache quantizations such as KVQuant, SKVQ, ZipCache, and CQ mentioned in the related work, yet the paper only compares against KIVI as the primary baseline. The absence of comprehensive comparisons with these recent methods makes it difficult to assess the relative merits of VidKV, especially since some of these methods also employ mixed-precision strategies or handle outliers differently. Furthermore, the paper lacks comparisons with other Video-LLM-specific compression techniques such as token pruning or merging methods (e.g., DyCoKe, HoliTom mentioned in related work), which could provide complementary or alternative solutions to the KV cache memory bottleneck and would help position VidKV's contributions more clearly within the broader landscape of VideoLLM efficiency research.

3. While the paper claims FFT adds "less than 5%" computational overhead, there is no detailed analysis of the actual inference latency of FFT in practice. The 1.58-bit ternary quantization is presented as enabling conversion of matrix multiplication to addition operations for computational efficiency, but no actual runtime measurements or throughput comparisons are provided to validate these theoretical benefits. In addition, it would be great to state the hardware support and implementation details, for instance, whether existing hardware accelerators can efficiently handle the mixed-precision scheme, FFT/IFFT operations during inference, and the ternary quantization format, which are critical factors for real-world deployment scenarios.

4. The results show notable performance degradation on certain benchmarks, particularly for the Qwen2.5-VL model on TempCompass and VideoDC when using 1.5-bit key and 1.58-bit value quantization, but the paper does not provide in-depth analysis of why these specific tasks are more sensitive to quantization. The VATEX captioning results (Table 4) show significant drops across all metrics for 1.x-bit configurations, suggesting the method may struggle with fine-grained generation tasks that require precise token representations.

**Questions:**

1. How the proposed method performs compared to additional methods, e.g., KVQuant, SKVQ, ZipCache, or CQ, or otation-based outlier smoothing methods (e.g., QuaRot, RotateKV)?

2. How the VidKV implement 1.58-bit quatization? What is the hardware support status for mixed-precision and ternary quantization formats?

3. What is the actual throughput improvement from converting multiplication to addition in 1.58-bit quantization?

---

### Note · Authors · 2026-01-18

I have read and agree with the venue's withdrawal policy on behalf of myself and my co-authors.